# Early Diagnosis of Cerebral Palsy in Low- and Middle-Income Countries

**DOI:** 10.3390/brainsci12050539

**Published:** 2022-04-23

**Authors:** Arrabella R. King, Mahmudul Hassan Al Imam, Sarah McIntyre, Catherine Morgan, Gulam Khandaker, Nadia Badawi, Atul Malhotra

**Affiliations:** 1Department of Paediatrics, Monash University, Melbourne, VIC 3168, Australia; arrabellaking@gmail.com; 2Central Queensland Public Health Unit, Central Queensland Hospital and Health Service, Rockhampton, QLD 4700, Australia; mahmudul.alimam@health.qld.gov.au (M.H.A.I.); gulam.khandaker@health.qld.gov.au (G.K.); 3Research Division, Central Queensland University, Rockhampton, QLD 4700, Australia; 4Cerebral Palsy Alliance Research Institute, Specialty of Child and Adolescent Health, Sydney Medical School, The University of Sydney, Sydney, NSW 2050, Australia; smcintyre@cerebralpalsy.org.au (S.M.); cmorgan@cerebralpalsy.org.au (C.M.); nadia.badawi@health.nsw.gov.au (N.B.); 5Discipline of Child and Adolescent Health, Sydney Medical School, The University of Sydney, Sydney, NSW 2050, Australia; 6Grace Centre for Newborn Intensive Care, The Children’s Hospital, Westmead, Sydney, NSW 2145, Australia; 7Monash Newborn, Monash Children’s Hospital, Melbourne, VIC 3168, Australia; 8The Ritchie Centre, Hudson Institute of Medical Research, Melbourne, VIC 3168, Australia

**Keywords:** cerebral palsy, cranial ultrasound, diagnosis, general movement assessment, Hammersmith infant neurological examination, low and middle income, magnetic resonance imaging, developmental delay

## Abstract

Cerebral palsy describes a group of permanent disorders of movement, motor function and posture that occur due to non-progressive insults to the developing brain. Most of the information concerning the early diagnosis of cerebral palsy originates from studies conducted in high-income countries. In this scoping review, we aimed to explore the tools used in low- and middle-income countries for the early diagnosis of cerebral palsy. A systematic search was conducted using OVID Medline and PubMed databases. “Early diagnosis” was defined as diagnosis prior to 12 months of age, and low- and middle-income countries were classified according to the World Bank classification system. We identified nine studies on the early diagnosis of cerebral palsy from low- and middle-income countries. The tools featured (n = number of studies) were: General Movement Assessment (6), neonatal magnetic resonance imaging (3), Hammersmith Neonatal Neurological Examination (2), Hammersmith Infant Neurological Examination (1) and cranial ultrasound (1). We found a paucity of published literature on the early diagnosis of cerebral palsy from low- and middle-income countries. Further research is needed to determine the tools that are accurate and feasible for use in low-resource settings, particularly since cerebral palsy is more prevalent in these areas.

## 1. Introduction

Cerebral palsy is defined as a group of permanent disorders of movement, posture and motor function that occur due to non-progressive insults within the immature brain [1]. It is recommended that cerebral palsy, or high risk of cerebral palsy, is diagnosed early in life to help to facilitate access to early targeted interventions and support services [2]. The 2017 guidelines published by Novak et al. [2] recommend two distinct pathways for the early diagnosis of cerebral palsy: infants with risk factors identified in the newborn period (newborn-detectable risk factors), such as prematurity, low birth weight or hypoxic ischaemic encephalopathy, should be assessed for cerebral palsy prior to five months of age, with detailed history and clinical examination, and a combination of neonatal magnetic resonance imaging (MRI), General Movement Assessment (GMA) and the Hammersmith Infant Neurological Examination (HINE); infants with no known neonatal risk factors and delayed motor milestones identified in the infantile period (infant-detectable risk factors) should be assessed for cerebral palsy between 5 to 24 months of age, using a combination of neonatal MRI, the HINE and standardised motor assessments, such as the Developmental Assessment of Young Children, the Alberta Infant Motor Scale, or the Neuro Sensory Motor Development Assessment [2].

High-risk infant follow-up clinics that incorporate the 2017 guidelines by Novak et al. [2] have been established in American and Australian healthcare centres in recent years [3,4,5]. These follow-up clinics have been shown to decrease the average age of cerebral palsy diagnosis and allow earlier access to targeted interventions and support for affected infants and families. To our knowledge, there is no current published literature on the establishment of high-risk infant follow-up clinics in low- and middle-income countries (LMIC), nor is it known whether it is appropriate to implement similar clinics in these areas. However, a recent study from Bangladesh (a typical LMIC) suggested that population-based surveillance of children with cerebral palsy enables early diagnosis and intervention [6].

Despite a higher prevalence of cerebral palsy reported by LMICs compared to high-income countries (HIC) [1,7,8,9,10], most of the literature concerning the early diagnosis of cerebral palsy originates from studies conducted in HICs. Tools used for the early diagnosis of cerebral palsy in HICs may not automatically translate into LMICs due to differences in population characteristics, healthcare system structure and utilisation, as well as issues with cost and resource availability. For example, preterm birth is underrepresented in children with cerebral palsy from LMICs compared to HICs due to a lack of access to perinatal care and survival bias in the living cohort [7]. In comparison, potentially modifiable perinatal risk factors, such as neonatal respiratory depression, encephalopathy and infection, are overrepresented in LMICs compared to HICs [7]. Further, the use of postnatal health services has been shown to be limited in LMICs due to socioeconomic status, educational level and geographical distance from services [11].

Several previous reviews have explored the tools used for the diagnosis of developmental delay in LMICs [12,13]. Developmental delay incorporates a broad range of neurodevelopmental conditions that may affect one or more of the developmental domains. It is likely that many children with cerebral palsy are incorporated within this diagnosis. Providing a diagnosis of cerebral palsy or high risk of cerebral palsy is advantageous over a diagnosis of developmental delay, as it helps to facilitate access to targeted interventions and support services [2].

To our knowledge, there is currently no published review on the tools used to diagnose cerebral palsy early within LMICs specifically. We feel this is an area that warrants research, given: the higher prevalence of cerebral palsy in LMICs compared to HICs [1,7,8,9,10]; the population and societal differences between LMICs and HICs, which may impede the ability for tools used in HICs to be successfully implemented into LICs [7,11], and the benefits that arise from diagnosing cerebral palsy early, including earlier access to targeted interventions and support [2]. We performed a scoping review to answer the following research question: what tools are currently being used to diagnose cerebral palsy early in LMICs? We aimed to systematically map the available research on the early diagnosis of cerebral palsy in LMICs, compare the tools used in LMICS to those used in HICs, and to identify gaps in knowledge and areas for further research.

## 2. Methods

A scoping review was conducted using the framework outlined by the Preferred Reporting Items for Systematic Reviews and Meta-analysis Protocols for Scoping Reviews (PRISMA-ScR) [14]. For the purpose of the review, the “early diagnosis” of cerebral palsy was defined as diagnosis prior to 12 months of age. Countries were classified as low and middle income using the World Bank classification system where low income, low-middle income, and upper-middle income countries are defined as countries with a gross national income per capita (in USD) of 1045 or less, 1046 to 4095, and 4096 to 12,696, respectively [15].

The scoping search was performed using OVID Medline and PubMed databases, and cross-checking the reference lists of relevant papers. The search strategy used in the databases is attached as an appendix (Appendix A). Two authors (AK and MA) performed the initial screen of articles by titles and abstracts. If there was a disagreement between the two authors, a third author (AM) was used to resolve conflicts. A single author (AK) performed the screen by full text. Peer-reviewed, original articles were included if they: included infants with no prior diagnosis of cerebral palsy; assessed the diagnosis of cerebral palsy prior to 12 months of age; were conducted in LMICs; were published in English; were published between November 2011 and November 2021, and included human participants. We excluded articles that assessed for cerebral palsy as an outcome of an intervention; studies conducted in HICs; published in languages other than English, and review articles, letters to editors, correspondence, study protocols or editorial comments. From the eligible articles identified in the search, we extracted data on the country in which the study was performed, the population studied, the follow-up rate of infants, and the tools used for early diagnosis. We summarised our findings in the form of a narrative synthesis.

## 3. Results

Figure 1 demonstrates the PRISMA flow chart for the scoping review. We identified nine articles on the early diagnosis of cerebral palsy in LMICs. Table 1 displays the studies identified within the search and Figure 2 provides pictorial representation of the tools used within these studies. Five of the nine studies were from upper-middle-income countries and four were from lower-middle-income countries. All identified studies were conducted in populations with newborn-detectable risk factors and none were conducted in populations with infant-detectable risk factors. Six of the nine studies utilised a neonatal intensive care follow-up program for recruitment of infants. The follow-up rates of neonates within these programs varied greatly from 50% to 95%. The guidelines by Novak et al. [2] were referenced in three of the five papers published after 2017 [16,17,18].

GMA was the most frequently studied tool for the early diagnosis of cerebral palsy and was included in six of the nine studies. Three studies reported on the accuracy of absent fidgety movements in predicting composite adverse outcomes, including cerebral palsy and developmental delay [17,22,24]. The sensitivity and specificity ranged from 62% to 89%, and 89% to 100%, respectively. Two studies reported on the accuracy of absent fidgety movements in diagnosing cerebral palsy specifically [16,21]. The sensitivity was 83% to 100%, and specificity was 96% to 100% [16,21]. One study reported on the accuracy of cramped synchronised movements in diagnosing cerebral palsy. The sensitivity and specificity were 82% and 99%, respectively [21]. 

The Hammersmith Neonatal Neurological Examination (HNNE) was featured in two of the nine studies. The sensitivity and specificity of the HNNE ranged from 50% to 69%, and 54% to 73%, respectively, for diagnosis of developmental delay and cerebral palsy [19,20]. The HINE featured in one of the nine studies. HINE scores of less than 67 were shown to have a sensitivity of 83% and specificity of 88% for diagnosing cerebral palsy [16]. 

Neonatal MRI featured in three of the nine studies. Two studies reported on the accuracy of MRI for predicting adverse developmental outcomes, including cerebral palsy [17,23]. The sensitivities and specificities ranged from 57% to 62%, and 79% to 90%, respectively. One study assessed the accuracy of MRI in predicting cerebral palsy specifically and found MRI to have sensitivity of 83% and specificity of 95% [16]. Neonatal cranial ultrasound featured in one study. It was shown to have a sensitivity and specificity of 31% and 96%, respectively, for diagnosing developmental delay and cerebral palsy [20]. Only one study included all three diagnostic tools recommended by Novak et al. [2] for the early diagnosis of cerebral palsy in infants with newborn-detectable risk factors [16]. The predictive abilities of all three tools in combination were not reported.

## 4. Discussion

To the best of our knowledge, this is one of the first scoping reviews to analyse the early diagnosis of cerebral palsy in LMICs exclusively. Current published reviews have largely focused on studies conducted in HICs, on singular diagnostic tools, or on developmental delay more broadly [12,13,25,26,27]. Overall, we identified few articles relating to the early diagnosis of cerebral palsy in LMICs. We identified a paucity of published research on the implementation of diagnostic tools into clinical practice and a lack of research concerning the early diagnosis of cerebral palsy in infants with infant-detectable risk factors. 

### 4.1. General Movement Assessment

GMA involves recording and analysing the movement patterns of infants during periods of calm wakefulness to assess for impairments in the immature nervous system [28]. Two abnormal movement patterns can predict the development of cerebral palsy: persistent cramped synchronised movements observed during the writhing phase from preterm age to nine weeks post-term age, and absent fidgety movements observed during the fidgety phase, most prevalent at 12 weeks and persistent until 16 to 20 weeks post-term age [28].

All identified studies on GMA assessed infants during the fidgety phase. Absent fidgety movements were frequently observed in infants who later went on to be diagnosed with cerebral palsy [16,18,21,24] and normal fidgety movements were frequently observed in infants who later went on to have neurotypical development [17,18,22,24]. Studies conducted in HICs on high-risk infants found that absent fidgety movements had a sensitivity and specificity ranging from 90% to 98% and 90% to 94%, respectively, for predicting cerebral palsy [29,30]. We found similar high accuracy for absent fidgety movements in LMICs. A 2016 Australian study found that in high-risk infants, abnormal or absent fidgety movements had a sensitivity and specificity of 54% and 97%, respectively, for diagnosing adverse neurological outcomes [29]. Our review identified a similar trend in LMICs, where studies assessing for composite adverse outcomes, including cerebral palsy and developmental delay, reported much lower sensitivities than those assessing for cerebral palsy alone. This suggests that absent fidgety movements may not be a suitable diagnostic tool to detect milder developmental impairments. 

One of the identified studies on GMA also assessed infants during the writhing phase. Dimitrijević et al. found that cramped synchronised movements had a sensitivity of 82% and specificity of 99% for diagnosing cerebral palsy in infants born preterm [21]. This study found the sensitivity of cramped synchronised movements to be lower when compared to absent fidgety movements. This fits with results from a 2018 systematic review, which demonstrated that cramped synchronised movements have a similar specificity but lower sensitivity for diagnosing cerebral palsy when compared to absent fidgety movements [31]. Although the sensitivity of cramped synchronised movements is lower, it has the advantage of being performed earlier in life while the infant is admitted to a neonatal care unit. Repeated assessments of cramped synchronised movements may, therefore, be more practical in areas with a neonatal care unit and result in less loss to follow up in LMICs, where the uptake of postnatal health services is limited [11]. Further research should focus on this area.

### 4.2. Hammersmith Neonatal Neurological Examination and Hammersmith Infant Neurological Examination

The Hammersmith neurological examinations are multi-item examinations that assess the neurological status of neonates and infants. There are two different examinations: the HNNE, a 34-item examination performed in the neonatal period [32] and the HINE, a 26-item examination performed between 2 to 24 months of age [33]. Global optimality scores are considered to be scores within the 90th centile range for healthy infants [25]. A HINE score of less than 57 at three months and less than 66 at 12 months is highly sensitive for diagnosing cerebral palsy [25]. 

The HNNE showed moderately low predictive ability in discerning neurodevelopmental delay or cerebral palsy in LMICs [19,20]. A 2016 Australian study found that suboptimal HNNE scores less than the 10th centile in moderate-to late-preterm infants were associated with higher odds of cognitive delay at 2 years, but there was little evidence of a relationship between HNNE scores and language or motor outcomes [34]. Venkata et al. compared the diagnostic ability of the HNNE performed early, pre-discharge from the neonatal intensive care unit to the HNNE performed at recommended age [19]. Variations in the timing of the examination did not appear to impact upon the accuracy of the test. Performing the HNNE earlier, pre-discharge resulted in fewer infants being lost to follow-up. Earlier timing of assessments may be more suitable in LMICs where follow-up is limited. 

The HINE was shown to have good predictive ability to diagnose cerebral palsy in term infants with hypoxic ischaemic encephalopathy, with a reported sensitivity of 83% and specificity of 88% [16]. This sensitivity is lower than that reported by a 2015 systematic review on HINE use in HICs [25]. The lower sensitivity may be due to issues with the follow-up of infants (only 80% of infants were followed up at two years of age), survival bias in LMICs, or differences in the population characteristics between infants from LMICs and HICs. 

Both Hammersmith neurological examinations rely on optimality scores and cut-offs to predict cerebral palsy. There are known differences between the optimality scores of healthy infants from LMICs and healthy infants from HICs [35,36]. This highlights that optimality scores from HICs will not automatically translate into LMICs, and demonstrates the need for further targeted research on HINE use in LMICs specifically. 

### 4.3. Neuroimaging

The neuroimaging techniques used to assist in the early diagnosis of cerebral palsy include neonatal MRI and cranial ultrasound. The most common neuroimaging finding in infants with cerebral palsy is white matter injury, followed by basal ganglia lesions, cortical and subcortical lesions, malformations and focal infarcts [37]. 

Medina-Alva et al. examined the cranial ultrasound findings at term age of very-low-birth-weight infants in Peru and reported that cranial ultrasound had a sensitivity of 31% and specificity of 96% for diagnosing developmental delay and cerebral palsy [20]. The sensitivity reported in this study is significantly lower than that reported by a meta-analysis (sensitivity of 74%) [26]. This may be secondary to Medina-Alva et al., using a composite outcome instead of assessing for cerebral palsy alone. It highlights that cranial ultrasound in isolation may not be a suitable tool for diagnosing milder development impairments. 

Neonatal MRI was found to be an accurate tool for predicting cerebral palsy in LMICs, with a sensitivity of 83% and specificity of 95% [16]. Similarly high accuracy has been demonstrated for neonatal MRI in HICs [38]. Cranial US has been shown to be less sensitive than MRI in predicting cerebral palsy in HICs [26,38]. While less accurate, cranial ultrasound may be more suitable for use in LMICs, as it is less expensive to perform and more accessible. Further studies should focus on the use of cranial ultrasound to assist in the diagnosis of cerebral palsy in LMICs.

### 4.4. Implementation into Clinical Practice

The studies identified within this scoping review used the diagnostic tools in a research capacity to define their accuracy and feasibility of use. We did not identify any published literature on the implementation of these tools into clinical practice within LMICs. Similarly, most of the studies focused on the use of a singular diagnostic tool in isolation. Only one study included all three of the recommended diagnostic tools for infants with newborn-detectable risk factors (GMA, HINE and neonatal MRI) [16]. Unfortunately, this study did not report on the accuracy of all three tools used in combination for diagnosing cerebral palsy. High-risk infant follow-up clinics that incorporate the guidelines by Novak et al. [2] have been established in HICs to aid in diagnosing cerebral palsy earlier [3,4,5]. While we did not identify any published literature on the establishment of similar clinics in LMICs, we did identify two study protocols from India, examining the use of community-based cerebral palsy detection programs [39,40]. This highlights that this is an area of emerging research within LMICs. Three of the five recent papers published after 2017 referenced the guidelines by Novak et al. [2], indicating that there is beginning to be uptake of these guidelines in LMICs. 

### 4.5. Populations Studied

As previously described, the guidelines by Novak et al. [2] included two arms: infants with newborn-detectable risk factors, such as prematurity, low birth weight or hypoxic ischaemic encephalopathy, and infants with infant-detectable risk factors, such as delayed motor milestones [2]. In this review, all of the identified studies were conducted on infants with newborn-detectable risk factors. Most of these studies used neonatal intensive care follow-up programs to recruit participants. The follow-up rates of these programs varied greatly, with some studies reporting follow-up rates as low as 50% [20,22]. The poor follow-up of infants was impacted upon by survival rates in LMICs. Other factors that may be contributing to low follow-up rates include geographical distance from services, health literacy, and socioeconomic status. It is likely that the poor follow-up of infants impacted upon the validity of the results reported in this review. Community-based early-detection models may help to improve the follow-up of these infants [39,40].

Half of all infants with cerebral palsy have risk factors identified in the infantile period [2]. None of the studies found in this review included infants within this group. This appears to be an area where there is a shortage of literature globally, with most studies from HICs also focusing exclusively on infants with newborn-detectable risk factors [2]. Infants with infant-detectable risk factors are likely to have less structured follow-up than those with newborn-detectable risk factors as they are often well during the newborn period, and risk factors are only identified later in life. How we assemble cohorts of these infants for diagnosis and research is an area that urgently requires more attention globally.

### 4.6. Developmental Delay

During the scoping review, we identified many studies on the early diagnosis of developmental delay in LMICs, but few on the early diagnosis of cerebral palsy specifically. An in-depth discussion of the tools used to diagnose developmental delay in LMICs is outside of the scope of this paper and has been discussed previously [12,13]. It is likely that many children with cerebral palsy are diagnosed under the umbrella term of developmental delay. The benefit of providing infants with a specific diagnosis of cerebral palsy is that it allows infants and families to receive targeted interventions and support services. The use of such intervention and support services appears to be limited in LMICs, likely due to factors including delayed diagnosis, service availability, cost, geographic distance from services, and education levels [41].

### 4.7. Limitations

This is a scoping review and, as such, our search might not have captured all published studies on the topic. Only papers published in English were included in the review, which may have excluded further studies relating to the research question. We did not perform statistical pooling of the results or meta-analysis, as there were no unifying outcomes.

## 5. Conclusions

There is an overall lack of published studies on the early diagnosis of cerebral palsy in LMICs, when compared to HICs. In infants with newborn-detectable risk factors, GMA, HINE and neonatal MRI have been shown to be accurate for diagnosing cerebral palsy early in LMICs; however, further studies utilising these tools are necessary. There is a paucity of published studies on the early diagnosis of cerebral palsy in infants with infant-detectable risk factors globally that urgently requires addressing. With further research on the early diagnosis of cerebral palsy in LMICs and the implementation of evidence-based medicine into clinical practice, we hope to see a reduction in the age of cerebral palsy diagnosis in LMICs, an increase in access to and utilisation of targeted therapies and support services, and ultimately, improved outcomes for these children.

## Figures and Tables

**Figure 1 brainsci-12-00539-f001:**
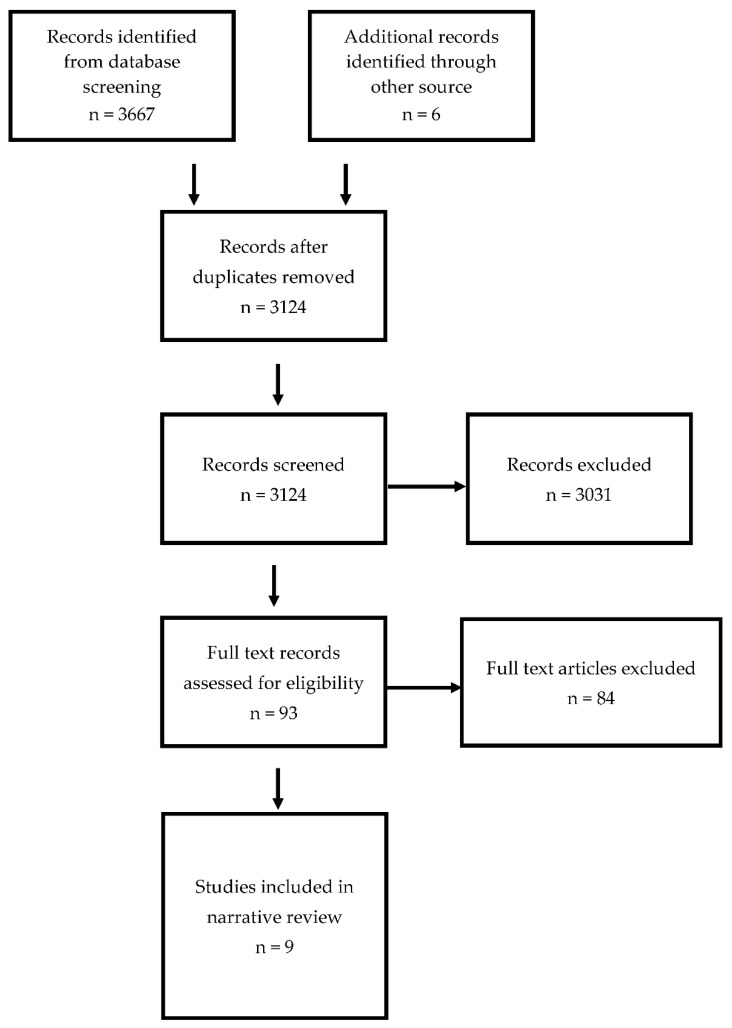
PRISMA flow diagram for scoping review.

**Figure 2 brainsci-12-00539-f002:**
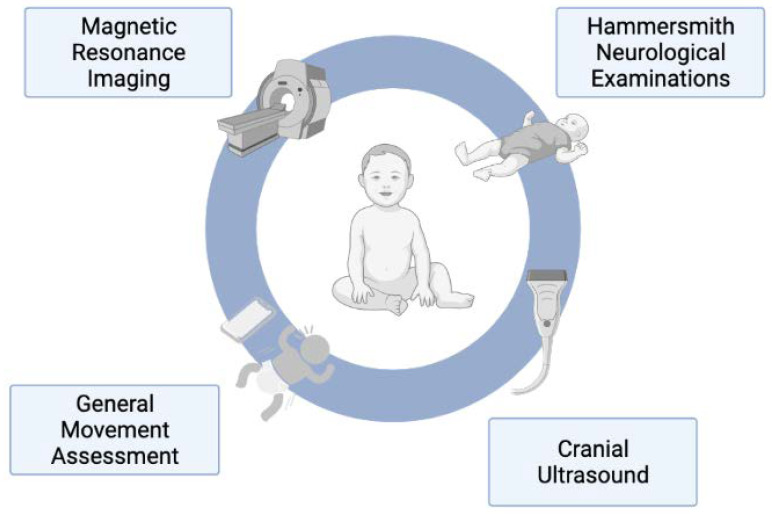
Tools used to diagnose cerebral palsy early in low- and middle-income countries.

**Table 1 brainsci-12-00539-t001:** Tools used for the early diagnosis of cerebral palsy in low- and middle-income countries.

Year	Author	Population	NICU Follow Up	Country (City/State) Income Level	Tools Used	Age of Assessment	Main Findings
2021	Aker [17]	Term/near term infants with moderate to severe hypoxic ischaemic encephalopathy	Yes	India (Vellore/Tamil Nadu) Lower-middle	MRI GMA	MRI at 5 days of life GMA at 10 to 15 weeks corrected age	MRI brain had a sensitivity of 62% and specificity of 90% for predicting adverse outcomes including cerebral palsy when compared to Bayley-III. Absent fidgety movements on GMA had a sensitivity of 60% and specificity of 89%.
2021	Apaydin [16]	Term infants with moderate to severe hypoxic ischaemic encephalopathy treated with hypothermia	No	Turkey (Ankara) Upper-middle	MRI GMA HINE	MRI 7 to 14 days after cooling GMA 12 weeks of age HINE 12 to 42 weeks	MRI had a sensitivity of 83% and specificity of 95% for predicting cerebral palsy diagnosis at 2 years of age when compared to the Bayley-II. Absent fidgety movements had a sensitivity of 83% and specificity of 100%, HINE had a sensitivity of 83% and specificity of 88%.
2020	Venkata [19]	Preterm infants and term infants admitted to the NICU with significant risk factors for cerebral palsy	Yes	India (Kerala) Lower- middle	HNNE	Early assessment performed after NICU care completed. Recommended assessment performed at 2 weeks of age (term) or 40 weeks corrected (preterm).	HNNE performed early had sensitivity of 64% and specificity of 73% for predicting neurodevelopmental disability, including cerebral palsy when compared to Development Assessment Scale for Indian Infants. HNNE performed at the recommended age had sensitivity of 50% and specificity of 77%.
2019	Einspieler [18]	Infants exposed to acute maternal zika infection	No	Brazil (Rio de Janeiro) Upper-middle	GMA	9 to 20 weeks corrected age	100% of infants with maternal zika-infection and microcephaly had abnormal or absent fidgety movements observed in infancy and developed bilateral spastic cerebral palsy at 12 months of age.
2019	Medina-Alva [20]	Preterm infants with birth weights between 500 and 2000 g	Yes	Peru (Lima) Upper-middle	Cranial ultrasound Head circumference HNNE	38 to 42 weeks corrected age	Cranial ultrasound had a sensitivity of 31% and specificity of 96% for detecting neurodevelopmental delay, including cerebral palsy at 24 months of age when compared to Mullen Scales of Early Learning. Abnormal HNNE score had a sensitivity of 69% and specificity of 54%. Microcephaly had sensitivity of 34% and specificity 97%.
2016	Dimitrijevic [21]	Preterm infants born < 37 weeks gestation	No	Serbia (Novi Sad) Upper-middle	GMA	4 to 12 weeks corrected age	Cramped synchronised movements had a sensitivity of 82% and specificity of 99% for predicting cerebral palsy at 24 months of age, when compared to paediatric examination. Absent fidgety movements had a sensitivity of 100% and specificity of 96%.
2015	Soleimani [22]	Infants with hypoxic ischaemic encephalopathy	Yes	Iran (Zanjan) Lower-middle	GMA	12 to 20 weeks corrected age	Absent fidgety movements had a sensitivity of 80% and specificity of 100% for detecting abnormal neurodevelopment including cerebral palsy, when compared to the Infant Neurological International Battery at 12 to 18 months of age.
2014	Lally [23]	Infants with hypoxic ischaemic encephalopathy	Yes	India (Kerala) Lower-middle	MRI	1 to 3 weeks chronological age	Abnormality on MRI imaging had a sensitivity 57% and specificity of 79% for diagnosing cerebral palsy or low Bayley III scores at 3 years of age, when compared to Bayley examination.
2011	Burger [24]	Preterm infants weighing </= 1250 g	Yes	South Africa (Western Cape) Upper-middle	GMA	12 weeks corrected age	Absent fidgety movements had a sensitivity of 89% and specificity of 89% for diagnosis neurodevelopmental delay and cerebral palsy at 12 months of age when compared to Pea Body Developmental Motor Scale and Alberta Infant Motor Scale.

CP—Cerebral palsy; GMA—General movement assessment; HINE—Hammersmith infant neurological examination; HNNE—Hammersmith neonatal neurological examination; MRI—Magnetic resonance imaging; NICU—Neonatal intensive care unit.

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
