# Peer review of "Early Diagnosis of Cerebral Palsy in Low- and Middle-Income Countries"

_brainsci, 2022, doi:10.3390/brainsci12050539_

Round 1
Reviewer 1 Report
The manuscript provides the assessment methods for early diagnosis of cerebral palsy in low and middle income countries. It’s well written and informative.
Please find my following comments:
- in the introduction, you mentioned that there is no clea assessment tools. Then, you conduct a review to summarize the used assessment tools.
- Please justify more the rational of your study.
- I don’t agree with the lines 91-105. It doesn’t seem sound. Finish the introduction with the study aim. These lines provide instructions for readers. Moreover, i did not hear about new methodology for scoping review.
- Please rewrite the conclusion to be more related to study results
Author Response
The manuscript provides the assessment methods for early diagnosis of cerebral palsy in low- and middle-income countries. It’s well written and informative.
Thank you for reviewing our manuscript and for your kind feedback.
Please find my following comments:
In the introduction, you mentioned that there is no clear assessment tools. Then, you conduct a review to summarize the used assessment tools.
We have removed that specific line and clarified that there are, to our knowledge, no published review articles on the tools used to diagnose cerebral palsy in low- and middle-income countries (Line 85).
Please justify more the rational of your study.
We have provided further justification for the rationale behind this scoping review (Lines 86 to 91).
I don’t agree with the lines 91-105. It doesn’t seem sound. Finish the introduction with the study aim. These lines provide instructions for readers. Moreover, i did not hear about new methodology for scoping review.
We have removed lines 91 to 105 from the manuscript.
Please rewrite the conclusion to be more related to study results
We have altered the conclusion to highlight the study results (lines 433 to 437).
Reviewer 2 Report
This is a review on available tools used for the early diagnosis of cerebral palsy in low-and middle-income countries (LMICs) as: The General Movement Assessment, neonatal magnetic resonance imaging, Hammersmith Neonatal Neurological Examination, Hammersmith Infant Neurological Examination and cranial ultrasound. The authors found a later diagnosis in LMICs and emphasize the need of further studies on precise and practical tools to use in these areas.
- The review is well structured, clear reporting in details tools for assessment of cerebral palsy, satisfying the topic purpose about the early diagnosis of developmental delay in LMICs and reporting a complete and appropriate references.
- About comments, the authors should deleted the author instructions about the paper:
lines 90-105, lines 130-134; complete the author contributions lines 378-385 and funding lines 386-389.
Author Response
This is a review on available tools used for the early diagnosis of cerebral palsy in low-and middle-income countries (LMICs) as: The General Movement Assessment, neonatal magnetic resonance imaging, Hammersmith Neonatal Neurological Examination, Hammersmith Infant Neurological Examination and cranial ultrasound. The authors found a later diagnosis in LMICs and emphasize the need of further studies on precise and practical tools to use in these areas.
Thank you for reviewing our paper.
The review is well structured, clear reporting in details tools for assessment of cerebral palsy, satisfying the topic purpose about the early diagnosis of developmental delay in LMICs and reporting a complete and appropriate references.
Thank you for your kind feedback.
About comments, the authors should deleted the author instructions about the paper:
lines 90-105, lines 130-134; complete the author contributions lines 378-385 and funding lines 386-389.
We have deleted the author instructions about the paper. We have completed the author contributions and funding lines (442 to 450)